# Palladium-Catalyzed Mizoroki–Heck and Copper-Free Sonogashira Coupling Reactions in Water Using Thermoresponsive Polymer Micelles

**DOI:** 10.3390/polym13162717

**Published:** 2021-08-13

**Authors:** Noriyuki Suzuki, Shun Koyama, Rina Koike, Nozomu Ebara, Rikito Arai, Yuko Takeoka, Masahiro Rikukawa, Fu-Yu Tsai

**Affiliations:** 1Department of Materials and Life Sciences, Faculty of Science and Technology, Sophia University, 7-1 Kioi-cho, Chiyoda-ku, Tokyo 102-8554, Japan; s-koyama-cm6@eagle.sophia.ac.jp (S.K.); r-koike-3gr@eagle.sophia.ac.jp (R.K.); n-ebara-397@eagle.sophia.ac.jp (N.E.); rikito5877@eagle.sophia.ac.jp (R.A.); y-tabuch@sophia.ac.jp (Y.T.); m-rikuka@sophia.ac.jp (M.R.); 2Institute of Organic and Polymeric Materials, National Taipei University of Technology, 1, Sec. 3, Chung-Hsiao E. Rd., Taipei 10608, Taiwan

**Keywords:** thermoresponsive polymer micelle, Mizoroki–Heck reaction, Sonogashira coupling, organic reactions in water

## Abstract

A few kinds of thermoresponsive diblock copolymers have been synthesized and utilized for palladium-catalyzed coupling reactions in water. Poly(*N*-isopropylacrylamide) (PNIPAAm) and poly(*N*,*N*-diethylacrylamide) (PDEAAm) are employed for thermoresponsive segments and poly(sodium 4-styrenesulfonate) (PSSNa) and poly(sodium 2-acrylamido-methylpropanesulfonate) (PAMPSNa) are employed for hydrophilic segments. Palladium-catalyzed Mizoroki–Heck reactions are performed in water and the efficiency of the extraction process is studied. More efficient extraction was observed for the PDEAAm copolymers when compared with the PNIPAAm copolymers and conventional surfactants. In the study of the Sonogashira coupling reactions in water, aggregative precipitation of the products was observed. Washing the precipitate with water gave the product with satisfactory purity with a good yield.

## 1. Introduction

The development of environmentally benign processes that enable organic syntheses to achieve the United Nations Sustainable Development Goals (SDGs) is an urgent subject. Transition metal-catalyzed chemical transformations are broadly utilized for producing fine chemicals; however, most catalytic reactions require organic solvents, which consequently results in an increase in the E-factor [1]. Conducting catalytic reactions in water is attractive to chemists who want to develop environmentally benign processes. In fact, many examples of palladium-catalyzed reactions conducted in water (or “on water”) have been reported over the past few decades [2,3,4,5,6,7,8,9,10,11,12,13]. The addition of surfactants to aqueous reaction mixtures causes the formation of oil/water (*o*/*w*) emulsions such that the organic reactions proceed in the micelle core. It is known that some chemical reactions can be accelerated in micellular systems [14,15,16,17]; however, organic reactions in *o*/*w* emulsions often require extraction processes using organic solvents to separate the products. Reducing the amount of the extraction solvent is an important subject for decreasing the E-factor of a process. Extraction processes might be more efficient if micelle formation can be “turned off” upon completion of the reaction. We envision that a thermoresponsive polymer micelle could be utilized for this purpose (Figure 1).

Poly(*N*-isopropylacrylamide) (PNIPAAm), which shows a lower critical solution temperature (LCST) at 32 °C in water, is known to be a thermoresponsive polymer and its applications in fields such as drug delivery systems and smart therapeutic materials have been studied extensively [18,19,20,21,22,23,24,25]. Thermoresponsive micelles that consist of PNIPAAm blocks have been vastly investigated, as has their utilization for therapeutic purposes [26,27,28,29,30]. For organic synthetic methods, there have also been many reports in which PNIPAAm was applied for organic reactions, as well as transition metal-catalyzed reactions. Many of these studies involve the use of a cross-linked PNIPAAm gel [31,32,33,34,35,36,37,38,39,40,41,42,43]; however, examples of thermoresponsive micelles applied for organic synthesis are still rare [44,45,46,47,48,49,50,51]. We previously reported the utilization of PNIPAAm block copolymers that form thermoresponsive micelles in water for organic synthesis. These micelles form at a temperature above 40 °C and dissociate at room temperature. We have tethered organocatalysts such as L-proline on the PNIPAAm block copolymers and demonstrated asymmetric cross-aldol reactions in water [44,46,47]. O’Reilly and coworkers also reported the use of PNIPAAm-based copolymer micelles bearing L-proline for asymmetric reactions in water [48]. We recently reported palladium-catalyzed Mizoroki–Heck reactions in water using these thermoresponsive polymer micelles [45] and showed that these reactions gave the products in high yields and with a good extraction efficiency. In the previous study, we reported that more efficient extraction was observed for aqueous solutions of the diblock copolymer poly(*N*-isopropylacrylamide)-*b*-poly(sodium 4-styrenesulfonate) (PNIPAAm-*b*-PSSNa, **NS**) compared to PNIPAAm*-b*-PEG, although the E-factor was still no less than 20. The extraction of more products from the aqueous reaction mixture with less organic solvent usage is important for elucidating an improved E-factor. Furthermore, the turnover number (TON) of palladium catalysts (2 mol %) was no more than 50, and reuse of the aqueous catalyst solutions was not achieved. Herein, we wish to report that Mizoroki–Heck reactions proceed in water using thermoresponsive micelles with a palladium catalyst **1** bearing 2,9-diphenyl-1,10-phenanthroline as a ligand. The TON reached 7800 due to high catalytic activity of **1**. In this study, we employ a new system of three diblock copolymers (**NA**, **DS** and **DA**), and examine the reactions with these copolymers as well as the extraction efficiency from the aqueous solutions (Figure 2). We also report palladium-catalyzed Sonogashira coupling reactions using these copolymers in water.

## 2. Materials and Methods

### 2.1. General

The preparation of copolymers was conducted under an argon atmosphere using standard Schlenk techniques unless otherwise mentioned. *N*-Isopropyl acrylamide (NIPAAm) was purchased from Kanto Chemical Co., Inc. and recrystallized from hexane/toluene prior to use. 2,2′-Azobis(isobutyronitrile) (AIBN), and dimethylacetamide (DMA) were purchased from Kanto Chemical Co., Inc. and used without further purification. Sodium dodecyl sulfate (SDS) and 4,4′-azobis(4-cyanovaleric acid) (V-501) were purchased from FUJIFILM Wako Pure Chemical Corporation and were used as received. *N*,*N*-Diethylacrylamide was purchased from Tokyo Chemical Industry Co., Ltd. and was distilled prior to use. Styrene was purchased from Tokyo Chemical Industry Co., Ltd., distilled, and kept under argon. Sodium 4-styrenesulfonate, 2-acrylamido-2-methylpropanesulfonic acid, dichlorobis(triphenylphosphine)palladium, 2,9-diphenyl-1,10-phenanthroline, iodobenzene, *n*-butyl acrylate, diisopropylethylamine, and α-methylstyrene were purchased from Tokyo Chemical Industry Co., Ltd. and were used as received. Other aryl halides, alkenes, ethynylarenes, and palladium catalysts were purchased and used as received. XPhos and Triton X-100 were purchased from Sigma-Aldrich Co. LLC. and used without further purification.

Palladium complex **1** was prepared from 2,9-diphenyl-1,10-phenanthroline and dichlorobis(acetonitrile)palladium according to the literature [52]. RAFT agent **2a** and **2b** were prepared according to the reported method in the literature [53,54]. Triethylammonium hypophosphite was prepared from triethylamine and hypophosphinic acid in toluene. The diblock copolymer **NS** was prepared as previously reported [45,55,56,57,58,59,60,61]. Dialysis was performed using Spectra/Por^®^ RC tubing (MWCO: 3.5kD). Deionized water was obtained on WE-200 (Yamato Scientific Co., Ltd., Tokyo, Japan). NMR spectra were recorded on JEOL ECA 500 and Bruker Avance III HD400 spectrometers. Gel permeation chromatography (GPC) was measured on PU-4580 and RI-4030 system (JASCO Corporation, Tokyo, Japan) equipped with Shodex GPC KD-802.5 and KD-804 columns (Showa Denko K.K., Tokyo, Japan) using *N*,*N*-dimethylformamide (DMF) (0.1 wt % LiBr) as an eluent. The molecular weight of the polymers was determined based on monodispersed poly(ethylene oxide) as standard. Dynamic light scattering (DLS) measurements were made with the DLS-8000 and ELSZ-2000ZS (Otsuka Electronics Co., Ltd., Osaka, Japan) instruments. Scanning transmission electron microscopy (STEM) was recorded with a S-8000 (Hitachi High-Tech Corporation, Tokyo, Japan) instrument. Transmittance was recorded on a Shimadzu UV-2550 instrument.

### 2.2. Preparation of the Homopolymer PNIPAAm

A thoroughly dried Schlenk tube (100 mL) was filled with argon. In this tube, RAFT agent **2b** (51 mg, 0.23 mmol), NIPAAm (0.54 g, 4.68 mmol), and AIBN (12 mg, 0.08 mmol) were dissolved in DMA (6 mL) and degassed in three freeze-pump-thaw cycles. The mixture was stirred at 60 °C for 24 h and the reaction mixture was poured into hexane/diethyl ether (75/75 mL) to precipitate a yellow solid. After the solvent was decanted, the yellow solid was dissolved in chloroform, the solution was collected, and the solvent was removed in a vacuum to leave the PNIPAAm homopolymer as a yellow solid (363 mg, 62%). The molecular weight was determined by ^1^H NMR spectroscopy. The DP (degree of polymerization) = 30, *M*_n_ = 3600 by ^1^H NMR. ^1^H NMR (D_2_O, Me_3_Si(C*H*_2_)_3_SO_3_Na, 500 MHz): δ 1.15 (C*H*_3_), 1.6 (C*H*_2_), 2.0–2.2 (C*H*), 3.89 (C*H*), 7.24–7.34 (br, *Ph*).

### 2.3. Preparation of the Copolymer PNIPAAm-b-PAMPSNa ***NA-T***

The obtained PNIPAAm (227 mg, 0.063 mmol) was added to a dried Schlenk tube, and sodium 2-acrylamido-2-methylpropanesulfonic acid (0.19 g, 0.83 mmol) and AIBN (6 mg, 0.04 mmol) were dissolved in dimethylsulfoxide (DMSO) (5 mL) in the tube. The mixture was degassed in three freeze-pump-thaw cycles. The tube was stirred at 65 °C for 17 h and the yellow mixture was purified by dialysis for 3 days. The dialyzed yellow solution was dried in a vacuum to produce the product polymer **NA-T** as a white solid (380 mg, 89%). The molecular weight was determined by ^1^H NMR spectroscopy. The degree of polymerization (DP) of the AMPSNa units = 9 and *M*_n_ = 5700 by ^1^H NMR. Our attempts to record GPC has been unsuccessful so far due to highly ionic property of the polymer. ^1^H NMR (D_2_O, Me_3_Si(C*H*_2_)_3_SO_3_Na, 500 MHz): δ 1.15 (C*H*_3_), 1.56 (C*H*_2_), 2.0–2.2 (C*H*), 3.4–3.6 (br, SC*H*_2_), 3.89 (C*H*), 7.24–7.34 (br, *Ph*).

### 2.4. Removal of the Ethyl Xanthogenate Terminus in the PNIPAAm-b-PAMPSNa ***NA-T***: Preparation of ***NA***

The PNIPAAm-*b*-PAMPSNa **NA-T** (308 mg, 0.067 mmol), triethylammonium hypophosphite (82 mg, 0.4 mmol), and V-501 (11 mg, 0.04 mmol) were dissolved in DMSO (5 mL) and the solution was degassed by the freeze-pump-thaw method. [62] The mixture was stirred at 80 °C for 3 h and additional V-501 (11 mg, 0.04 mmol) was added to the solution. After the mixture was stirred at 60 °C for 17 h, the yellow solution was dialyzed. The resultant colorless solution with white precipitate was dried in vacuo to obtain the product **NA** as a white solid (281 mg, 76%). The molecular weight was determined by ^1^H NMR spectroscopy. DP of the PNIPAAm segment was 30, while PAMPSNa segment was 9, *M*_n_ = 5600 by ^1^H NMR. ^1^H NMR (D_2_O, Me_3_Si(C*H*_2_)_3_SO_3_Na, 500 MHz): δ 1.15 (C*H*_3_), 1.56 (C*H*_2_), 2.0–2.2 (C*H*), 3.4–3.6 (br, SC*H*_2_), 3.89 (C*H*), 7.24–7.34 (br, *Ph*).

### 2.5. Preparation of the Homopolymer PDEAAm

In a dried Schlenk tube (25 mL), RAFT agent **2a** (92 mg, 0.36 mmol), *N*,*N*-diethylacrylamide (916 mg, 7.2 mmol), and AIBN (16 mg, 0.10 mmol) were dissolved in DMA (7 mL). The mixture was degassed in three freeze-pump-thaw cycles and was stirred at 60 °C for 24 h. The solution was poured into hexane (400 mL) and the yellow precipitate was dissolved in chloroform. The volatiles were removed in vacuo to produce PDEAAm as a yellow solid (529 mg, 53%). The average molecular weight of the polymer was determined as *M*_n_ = 2300, *M*_w_/*M*_n_ = 1.20 by gel permeation chromatography (GPC) analysis using poly(ethylene oxide) as a standard and *M*_n_ = 3200 (*DP* = 23) as per ^1^H NMR. ^1^H NMR (D_2_O, Me_3_Si(C*H*_2_)_3_SO_3_Na, 500 MHz): δ 1.10–1.25 (C*H*-C*H*_3_), 1.5–1.8 (C*H*_2_), 2.5–2.8 (C*H*), 3.2–3.5 (NC*H*_2_), 7.24–7.34 (br, *Ph*).

### 2.6. Preparation of the Copolymer PDEAAm-b-PSSNa ***DS-T***

In a thoroughly dried Schlenk tube, the obtained PDEAAm (255 mg, 0.08 mmol) was dissolved in DMSO (5 mL) and sodium 4-styrene sulfonate (167 mg, 0.8 mmol) and AIBN (5.2 mg, 0.032 mmol) were added. The mixture was degassed in 3 freeze-pump-thaw cycles and stirred at 65 °C for 24 h. The yellow mixture was dialyzed for 2 days. The volatiles were removed from the dialyzed mixture in vacuo to afford the title compound as a yellow solid (415 mg, 97%). ^1^H NMR (D_2_O, Me_3_Si(C*H*_2_)_3_SO_3_Na, 500 MHz): δ 1.10–1.25 (C*H*-C*H*_3_), 1.5–1.8 (C*H*_2_), 2.5–2.8 (C*H*), 3.2–3.5 (NC*H*_2_), 7.24–7.34 (br, *Ph*), 7.6–7.8 (C_6_*H*_4_).

### 2.7. Removal of Trithiocarbonate Terminus from DS-T; Synthesis of ***DS***

In a thoroughly dried Schlenk tube (100 mL), the prepared **DS-T** (769 mg, 0.138 mmol), triethylammonium hypophosphite (172 mg, 0.833 mmol), and V-501 (28 mg, 0.1 mmol) were dissolved in DMSO (8 mL). The mixture was degassed by 3 freeze-pump-thaw cycles and heated at 80 °C for 24 h. The yellow solution was dialyzed for 25 h. The volatiles were removed in vacuo to give the title compound as a white solid (662 mg, 89%). DP was determined by ^1^H NMR (*m* = 23, *n* = 9, *M*_n_ = 4900). ^1^H NMR (D_2_O, Me_3_Si(C*H*_2_)_3_SO_3_Na, 500 MHz): δ 1.10–1.25 (CH-C*H*_3_), 1.5–1.8 (C*H*_2_), 2.5–2.8 (C*H*), 3.2–3.5 (NC*H*_2_), 7.24–7.34 (br, *Ph*), 7.6–7.8 (C_6_*H*_4_).

### 2.8. Preparation of the Diblock Copolymer Poly(DEAAm-b-AMPSNa) ***DA-T***

The diethylacrylamide homopolymer was prepared as described above (*DP* = 34). The obtained PDEAAm homopolymer (428 mg, 0.12 mmol), sodium 2-acrylamide-2-methylpropane sulfonate (444 mg, 0.6 mmol), and AIBN (9 mg, 0.055 mmol) were dissolved in DMSO (7 mL) and degassed in three freeze-pump-thaw cycles. The mixture was stirred at 65 °C for 17 h. The yellow mixture was dialyzed, and the volatiles were removed in vacuo. The title compound was obtained as a white solid (380 mg, 78%). ^1^H NMR (D_2_O, Me_3_Si(C*H*_2_)_3_SO_3_Na, 500 MHz): δ 1.10–1.25 (C*H*-C*H*_3_), 1.5 (C*H*_3_), 1.6–1.8 (C*H*_2_), 2.5–2.8 (C*H*), 3.2–3.5 (NC*H*_2_ + SC*H*_2_), 7.24–7.34 (br, *Ph*).

### 2.9. Removal of Trithiocarbonate Terminus from ***DA-T***; Synthesis of ***DA***

The obtained polymer **DA-T** (444 mg, 0.12 mmol) was dissolved in DMSO (6 mL), and triethylammonium hypophosphite (112 mg, 0.54 mmol) and V-501 (15 mg, 0.054 mmol) were added to this solution. The mixture was degassed by the freeze-pump-thaw method and stirred at 80 °C for 3 h. Additional V-501 (15 mg, 0.054 mmol) was added to the solution. After the mixture was stirred at 60 °C for 17 h, the yellow solution was dialyzed. The resultant colorless solution with a white precipitate was dried in vacuo to afford the product as a white solid (281 mg, 76%). DP was determined by ^1^H NMR (*m* = 34, *n* = 3, *M*_n_ = 5100). ^1^H NMR (D_2_O, Me_3_Si(C*H*_2_)_3_SO_3_Na, 500 MHz): δ 1.10–1.25 (C*H*-C*H*_3_), 1.5 (C*H*_3_), 1.6–1.8 (C*H*_2_), 2.5–2.8 (C*H*), 3.2–3.5 (NC*H*_2_ + SC*H*_2_), 7.24–7.34 (br, *Ph*).

### 2.10. Mizoroki–Heck Reactions in Water Using the Copolymers, Initial Study

The typical procedure for Mizoroki–Heck reactions in water using the thermoresponsive micelles is as follows. In a test tube with a screw cap, the copolymer (40 mg) was dissolved in deionized water (4 mL) and the solution was stirred. With this solution, iodobenzene (102 mg, 0.5 mmol), *n*-butyl acrylate (128 mg, 1.0 mmol), PdCl_2_(PPh_3_)_2_ (7.0 mg, 0.01 mmol), and diisopropylethylamine (129 mg, 1.0 mmol) were added and the mixture was stirred at 70 °C for 48 h. The grayish turbid suspension was cooled in an ice bath and then diethyl ether (3 mL) was added and stirred for 1 h. The organic layer was taken up and extracted again with diethyl ether (3 mL) until the product was not detected by thin layer chromatography. The extract was analyzed by gas chromatography using tetradecane as an internal standard to determine the GC yield.

### 2.11. Mizoroki–Heck Reactions in Water Using the Copolymers Catalyzed by ***1***

Typically, in a test tube with a screw cap, the copolymer (20 mg) was dissolved in deionized water (1 mL) and the solution was stirred. With this solution, iodobenzene (102 mg, 0.5 mmol), *n*-butyl acrylate (128 mg, 1.0 mmol), and tri-*n*-butylamine (185 mg, 1.0 mmol) were added and the mixture was stirred. Meanwhile, palladium complex **1** (2.4 mg, 0.005 mmol) was dissolved in *N*-methyl-2-pyrrolidone (400 μL) in a vial. With this yellow solution, hydrazine monohydrate was added (1.6 mg, 0.032 mmol) and stirred for 10 s. This solution (40 μL) was added to the test tube and the mixture was stirred at 70 °C for 48 h. The turbid suspension was cooled in an ice bath and then ethyl acetate (0.4 mL) was added and vigorously stirred, then centrifuged at 1000 rpm for 10 min (180× *g*). The organic layer was taken up and extracted again with ethyl acetate (0.4 mL) until the product was not detected by thin layer chromatography. The extract was analyzed by gas chromatography using tetradecane as an internal standard to determine the GC yield (97%).

### 2.12. Evaluation of Extraction Efficiencies of Mizoroki–Heck Product ***5aa***

Typically, in a test tube with screw cap, the block copolymer **DS** (40 mg) was dissolved in deionized water (4 mL) and the solution was stirred for 0.5 h at room temperature. With this solution, *n*-butyl cinnamate (**5aa**, 102 mg, 0.5 mmol) was added and the mixture was stirred at 70 °C for 1 h. Ethyl acetate (1 mL) was added to this mixture, and this mixture was shaken at 120 rpm in a shaking apparatus at 0 °C for 0.5 h. The mixture was allowed to stand still for 0.5 h at room temperature, and then the organic layer was taken up and analyzed by gas chromatograph using tetradecane as an internal standard.

### 2.13. Sonogashira Coupling Reactions in Water Using the Copolymers

The typical procedure for Sonogashira coupling in water is as follows. In a dried test tube with a screw cap, copolymer **NS** (20 mg) was dissolved in deionized water (2 mL). Then, 4-iodoanisole (117 mg, 0.5 mmol), ethynylbenzene (77 mg, 0.75 mmol), PdCl_2_(PPh_3_)_2_ (7.0 mg, 0.01 mmol), and triethylamine (110 mg, 1.0 mmol) were added, and the mixture was then stirred at 70 °C for 24 h. The brown turbid solution was cooled in an ice bath while stirring until the supernatant became clear. The precipitated brown solid was collected by filtration when possible. Otherwise, the mixture was extracted with ethyl acetate and purified by column chromatograph on silica gel (hexane/ethyl acetate = 4/1). The solid was characterized by ^1^H NMR in CDCl_3_.

## 3. Results and Discussion

### 3.1. Preparation and Temperature-Dependent Properties of the Diblock Copolymers

We previously reported the use of the thermoresponsive diblock copolymer poly(*N*-isopropylacrylamide-*b*-sodium 4-styrenesulfonate) (PNIPAAm-*b*-PSSNa, **NS**) in palladium-catalyzed reactions in water. [45] In this study, we also employed poly(*N*,*N*-diethylacrylamide) (PDEAAm) as a thermoresponsive block that shows LCST at 35–40 °C in water [63]. We envisioned that the enhanced hydrophobic properties of the *N*,*N*-diethylamide moieties compared with PNIPAAm may facilitate intake of organic substrates into the micelle core. In addition, poly(sodium 2-acrylamide-2-methylpropane sulfonate) (PAMPSNa) was used as a hydrophilic segment as the amide moiety can be expected to be more hydrophilic than PSSNa that bears aromatic rings.

Thus, in addition to the previously reported **NS**, three diblock copolymers were prepared, i.e., PNIPAAm-*b*-PAMPSNa **NA** [64,65,66], PDEAAm-*b*-PSSNa **DS** [67], and PDEAAm-*b*-PAMPSNa **DA**. All polymers were synthesized by a reversible addition-fragmentation chain-transfer (RAFT) polymerization technique. For example, DEAAm was polymerized in the presence of RAFT agent **2a** to give a PDEAAm homopolymer, followed by polymerization of AMPSNa on PDEAAm to give a PDEAAm-*b*-PAMPSNa diblock copolymer **DA-T** which has a trithiocarbonate terminus. Removal of the sulfur-containing moieties at the termini afforded **DA** (Scheme 1). Typical examples of the polymerization degree as determined by ^1^H NMR are shown in Scheme 1. To the best of our knowledge, diblock copolymer **DA** has not been reported to date, although the random copolymer was reported [68].

The temperature-dependent transmittance of the aqueous solution of the copolymers indicated their thermoresponsive behavior (Figure 3). Copolymer **NA**, which has a PNIPAAm block as the thermoresponsive segment, showed LCST at 30–35 °C, while copolymers **DS** and **DA**, which have a thermoresponsive PDEAAm block, showed LCST at 35–40 °C.

Dynamic light scattering (DLS) analysis of aqueous solutions of the copolymers indicated that the particle sizes at 50 °C were significantly larger than those at 30 °C, showing the thermoresponsive formation of copolymer micelles in water (Figure 4). It is noteworthy that the copolymers **NA** and **DA** showed smaller particle sizes as ca. 10 nm at 30 °C, compared with **NS** and **DS** (40 nm and 150 nm, respectively). It is possibly because **NA** and **DA** are well dispersed at 30 °C, owing to more hydrophilic PAMPSNa segments. On the other hand, **DS** and **DA** formed larger particles at 50 °C (270–380 nm) than those formed by **NA**. These results might suggest that more hydrophobic PDEAAm segments tend to form more aggregated micelles above LCST.

### 3.2. Palladium-Catalyzed Mizoroki–Heck Reaction Using the Copolymers

We studied the Mizoroki–Heck reaction in water using the thermoresponsive polymer micelles. We previously reported that copolymer **NS** promoted the Mizoroki–Heck reaction in water with PdCl_2_(PPh_3_)_2_ as a catalyst precursor [45]. In this study, the three other copolymers were also investigated. Initially, we compared these polymers under our previous reaction conditions (Table 1). The copolymer was dissolved in water at room temperature and to this aqueous solution was added the substrates, base, and the palladium catalyst. As the mixture was heated at 70 °C and stirred, the mixture became opaque. After stirring, the reaction mixture was cooled to room temperature whereupon the solution became clear. The product was extracted and the yield was determined by gas chromatography or ^1^H NMR.

The reaction using copolymer **NA** delivered a moderate yield (entry 3), whereas those with **DS** and **DA** resulted in satisfactory results (entries 4 and 6). These results were superior to those obtained by using conventional surfactants such as sodium dodecyl sulfate (SDS) and Triton X-100 (entries 7 and 8), as well as those obtained without any surfactants (only water, entry 9) under the same reaction conditions. The aqueous solution of **NA** was much more viscous than other polymer solutions, and the lower yield obtained with **NA** might be due to slow diffusion of the substrates. In these reactions, 2 mol % of palladium complex PdCl_2_(PPh_3_)_2_ was required to complete the reactions. A decrease of the Pd load to 0.5 mol % resulted in lower yields (entries 2 and 5).

In our previous report, we examined a range of Pd complexes and concluded that PdCl_2_(PPh_3_)_2_ was optimal. In this study, we employed the Pd complex of NNC-pincer ligand **1** as a catalyst precursor [69,70,71,72]. Uozumi and coworkers reported excellent catalytic activity of ligand **1** for various reactions [52,73,74,75]. We adopted **1** for the Mizoroki–Heck reactions using **NS** (Table 2). When only **1** was added to the system, the reaction resulted in a moderate yield (entry 1). We considered that the Pd(II) species was not effectively reduced in the reaction system. Hydrazine hydrate was then mixed with **1** prior to use in a small amount of *N*-methyl-2-pyrrolidone (NMP) and the NMP solution was added to the reaction mixture. Addition of hydrazine improved the catalytic activity remarkably, and the reactions gave the product in quantitative yields with less Pd load (0.1 mol%), in shorter reaction time (entry 3). Even the use of 0.01 mol % Pd was sufficient to achieve a good yield, and the TON reached more than 6000, although the reaction was somewhat irreproducible. Thus, we decided to conduct further studies with 0.1 mol % of the Pd catalyst.

These results led us to investigate the reaction with various substrates with other copolymers using **1** in water (Table 3). Copolymers **DS** and **DA** delivered similar results to **NS**, whereas the use of **NA** resulted in lower yield (entries 1–4). Again, it is presumed that the low yield resulted from the high viscosity of the **NA** solution. Most iodoarenes reacted with *n*-butyl acrylate to give the products in good to excellent yields in the presence of 0.1 mol % ligand **1** and hydrazine regardless of the presence of electron-donating or electron-withdrawing groups on the aromatic ring (entries 6–10), whereas bromobenzene resulted in a low yield (entry 5). Other acrylic ester and amide gave the corresponding coupling products in good yield (entries 11–12). On the other hand, the use of styrene derivatives as a coupling partner gave lower yields (entries 13–15). In the attempts on heteroarenes such as iodopyridines and 2-iodothiophene, most of the starting materials remained intact (entries 16–18). Hex-1-ene gave a trace amount of the products as isomeric mixtures (entry 19).

### 3.3. Reuse of the Aqueous Solution and Formation of Palladium Nanoparticles (PdNPs)

The aqueous solution after the reaction between iodobenzene and *n*-butyl acrylate was reused for further reactions. The substrates and bases were added and the mixture was stirred at 100 °C for 24 h. To our delight, the second reaction gave the product in 95% yield, and the third run delivered a 62% yield. Observation of the aqueous solution after the reactions by scanning transmission electron microscopy (STEM) indicated the formation of nanoparticles of palladium with diameters of 30–80 nm (Appendix A). Uozumi and coworkers proposed that the palladium atoms in **1** form nanoparticles (NPs) in the reaction solution and that single atoms liberated from the particles catalyze the reactions [52,73]. In this reaction, it is likely that PdNPs generated by the reduction with hydrazine are encapsuled and protected by the polymers in water, at room temperature. Gradual growth of the particle might cause the decrease of the yield.

### 3.4. Extraction Efficiency

One of the problems in organic reactions in aqueous media when using micelles is the separation of the products from the reaction mixture. The products are commonly extracted with considerable amounts of organic solvents. Thus, reducing the amount of extraction solvents is important to achieve an environmentally benign system. In our previous study, we examined the efficiency in extracting the Mizoroki–Heck product **5aa** from the aqueous solution of **NS** [45]. We herein also studied the other polymers.

Model aqueous mixtures were prepared that consisted of surfactants (1 wt %) and **5aa** in water and these were stirred at 70 °C for 1 h. The mixtures were then cooled and extracted once at 0 °C with 1 mL of an extraction solvent such as diethyl ether or ethyl acetate. The results are summarized in Table 4. The extraction efficiency was estimated on the basis of the amount of recovered **5aa** within a given period of extraction time. Overall, the extraction with ethyl acetate recovered more **5aa** than with diethyl ether.

In the absence of any surfactants, 50–60% of **5aa** was recovered within 30 min (entry 1). To our delight, more **5aa** was recovered from most of the copolymer solutions (**NA**, **DS** and **DA**). This is probably due to salting out effect by sulfonate ions. Note that the DEAAm copolymer solutions of **DS** and **DA** showed better results compared with those of the NIPAAm copolymer **NS**. This is presumably because of the more hydrophobic nature of *N*,*N*-diethylamide moieties. The extraction efficiency from the suspension of SDS was comparable to that from water (entry 6), although the palladium-catalyzed reaction gave the product in lower yield in SDS suspension under the present reaction conditions, as we previously reported [45]. We conducted the same study using the reaction mixture with **DA** (entries 7–8). Single extraction with 1 mL of ethyl acetate gave 83% recovered product **5aa** (entry 7), and 0.5 mL of ethyl acetate gave 68% recovered product (entry 8). The calculated E-factors in these entries were 12.6 and 8.9, respectively.

### 3.5. Sonogashira Coupling Reaction

Sonogashira coupling reaction is a powerful tool for the construction of alkynylarene scaffolds. The reaction was originally promoted by palladium catalysts and copper salts as co-catalysts. Recently, it was found that the reactions can proceed under copper-free conditions [76,77], especially in aqueous media [78,79]. We herein examined copper-free Sonogashira coupling reactions in the thermoresponsive micelle system.

First, palladium catalysts were examined in the absence of copper salts for the reaction between 4-iodoanisole (**3d**) and phenylacetylene (**6a**) using **NS** as a copolymer surfactant (Table 5). Among a selection of palladium(II) catalyst precursors (entries 1–5), PdCl_2_(PPh_3_)_2_ gave the product **7da** in 86% yield, even in the absence of Cu salt (entry 2), whereas the yield reached 99% when CuI was added (entry 11). Combination of Pd(II) species and phosphine ligands were also studied (entries 6–9). Although triphenylphosphine and tricyclohexylphosphine gave low to moderate yields (entries 6 and 7), the addition of XPhos, which is known to be effective for copper-free Sonogashira coupling [80,81,82], increased the yield (entries 8 and 9). Palladium complex **1**, which was highly active for Mizoroki–Heck reactions, was not so active in this reaction (entry 10). Nickel complexes showed low activity (entries 12 and 13). Thus, we selected the conditions described in entry 2 for further studies on the effects of surfactant copolymers. The results are summarized in Table 6.

The reaction between 4-iodoanisole and phenylacetylene in water using various polymers gave the Sonogashira product **7da** with moderate to good yields. Reactions with the copolymers **NS**, **NA**, **DS**, and **DA** gave slightly higher yields than with conventional surfactants. Interestingly, when polymer **NS** was applied (Table 6, entry 2), the product **7da** precipitated at the bottom of the reaction vessel as aggregated chunks (Figure 5). Before the reaction, added substrates were biphasic and separated. The mixture was finely suspended with the formation of micelles during the reaction at 70 °C. The mixture was cooled after the reaction with stirring, then the aggregated precipitates could be easily taken up by filtration and washed with water. Analysis of the washed solid by ^1^H NMR spectroscopy indicated that it was an almost pure product (84% yield, see the supporting information for the NMR spectrum). The polymer **NS** gave better precipitates, whereas other copolymers and surfactants afforded sticky solids that made separation more difficult. Although the procedure was somewhat irreproducible with respect to the formation of the aggregated chunks, the calculated E-factor in this method was 1.8 and the process might provide an advantageous method for product separation from the reaction mixture.

We then investigated the scope and limitations of the substrates in Sonogashira coupling reactions using **NS** as thermoresponsive micelles (Table 7). Aryl iodides bearing either electron-donating or electron-withdrawing groups gave the coupled products with good yields (entries 1 and 3), although aryl bromide gave the product in a low yield with PdCl_2_(PPh_3_)_2_ (entry 2). Pd(OAc)_2_ with XPhos, on the other hand, successfully promoted the reaction of aryl bromides (entries 6, 8–10, 12), except for the reaction between 2-bromo-6-methylpyridine and (*tert*-butyldimethylsilyl)acetylene (entry 11). The reaction of aryl chloride was disappointing (entry 7). It was demonstrated that Pd(OAc)_2_/XPhos was effective as a catalyst in water for various aryl bromides, including alkenyl bromide (entry 13), and terminal alkynes. [82] On the contrary to the Mizoroki–Heck reactions, heteroarenes such as 2-iodothiophene and 2-bromopyridine gave the coupling products in moderate to good yields (entries 5, 9 and 11).

## 4. Conclusions

We have prepared thermoresponsive diblock copolymers that consist of poly(*N*-isopropylacrylamide)/poly(*N*,*N*-diethylacrylamide) as thermoresponsive segments and poly(sodium 4-styrenesulfonate)/poly(sodium 2-acrylamido-methylpropanesulfonate) as a hydrophilic segment. These copolymers formed micelles at 50 °C in water and they dissolved at room temperature. Palladium-catalyzed Mizoroki–Heck reactions proceeded in water using these thermoresponsive micelles. In particular, the palladium complex of NNC-pincer ligand **1** achieved a TON result as high as 7800. Copper-free Sonogashira coupling reactions were promoted in water using the copolymers. We achieved to decrease the use of organic extract by using the thermoresponsive polymer micelles.

## Data Availability

The data presented in this study are available on request from the corresponding author.

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
