# Peer review of "Palladium-Catalyzed Mizoroki–Heck and Copper-Free Sonogashira Coupling Reactions in Water Using Thermoresponsive Polymer Micelles"

_polymers, 2021, doi:10.3390/polym13162717_

Round 1

Reviewer 1 Report

This manuscript described synthesis of a series of block copolymers and their applications in aqueous Pd-catalyzed coupling reactions. The authors provided detailed procedures for all experiments and measurements, demonstrating a high level of scientific soundness. The investigations were performed to demonstrate the efficiency of using these copolymers in aqueous coupling reactions. This method can be used for a wide scope of substrates and the yields were optimized to be higher than 80% for most reactions. Moreover, the diblock copolymer solution can be reused after extraction, which contributed to the sustainability of this method. Overall, this paper shows a significant advance in the field of polymer science and its applications. I would recommend this paper to be accepted in its current form.  

Author Response

We thank the reviewer for his/her favorable comment.

Reviewer 2 Report

These are cleverly devised experiments.   It is a modern application of the extensively studied reactions under micellar conditions.  I might comment that the procedure does not preclude the use of organic solvents as the workup involves ethyl acetate extraction, and this takes the edge off the claim of "aqueous"  conditions. Nonetheless, the the work is new and interesting and worthy of publication.

Author Response

We thank the reviewer for his/her comment. As the reviewer mentioned, use of extraction solvents can be problematic for aqueous reaction, although most reported “aqueous” organic reactions involve extraction processes. The aim of our study is to decrease or exclude any use of organic solvents by using thermoresponsive micelle systems. We believe that we could achieve this goal to some extent in this study.

Reviewer 3 Report

The manuscript is well written and has covered the most significant papers on the palladium-catalyzed Mizoroki-Heck reactions and Sonogashira reactions in water. However, there are various limitations of this article. There are no obvious advantages using these thermoresponsive polymer micelles. The use of micelles in water with ppm level of Pd-catalyst has shown to work with a much broader scope of the substrate including aryl halide as well as heteroaryl halide with both electron-withdrawing and electron-donating group. The substrate scope of the paper is very limited. Due to the lack of novelty, I would recommend the author to submit the paper elsewhere or resubmit after the improvement of the reactions.

  • Why aryl bromide gave a good yield for Sonogashira Coupling compared to the Mizoroki-Heck reaction?
  • The use of hydrazine for the reduction of the Pd-catalyst before reaction gave a good yield in a short time. Could you reduce Pd-catalyst in the reaction using other reducing agents?
  • The statement in lines 327-329 “Most iodoarenes reacted with acrylate esters or acrylamides to give the products in good to excellent yields in the presence of 0.1 mol% 1 and hydrazine regardless of the presence of electron-donating or electron-withdrawing groups on the aromatic ring (entries 6-9)” is not suitable. Only one substrate with methoxy group and ketone was used. Please include a couple more substrates with electron-donating and electron-withdrawing groups.
  • The reaction temperature for the Mizoroki-Heck reaction is high. The ppm level of Pd is shown to work in water conditions at rt-45 °C. Does the reaction work at a low temperatures?
  • Similarly, the reaction temperature for Sonogashira coupling is high given that the catalysis in water using Pd-catalyst has been shown to work at lower temperatures i.e., 45 °C with broader substrate scope.
  • The substrate scope of the reaction is limited to the aryl substrate. Could you please use some of the heteroaryl-substrate? The is one heteroaryl substrate used for Sonogashira coupling with lower yield i.e., 33 % yield.

Author Response

We thank the reviewer for valuable comments. The advantages of thermoresponsive polymers should have been described more clearly. Conclusion was amended so that it is more emphasized.

  • Why aryl bromide gave a good yield for Sonogashira Coupling compared to the Mizoroki-Heck reaction?

It was reported that XPhos was excellent ligand for copper-free Pd-catalyzed Sonogashira coupling so that aryl bromide can join the reactions. Because there was no such a convenient ligand for Mizoroki-Heck reactions under our conditions, aryl bromides gave poor yield. We inserted the reference [82] into the paragraph.

  • The use of hydrazine for the reduction of the Pd-catalyst before reaction gave a good yield in a short time. Could you reduce Pd-catalyst in the reaction using other reducing agents?

Indeed, we employed other reductants such as hydroquinone and ascorbic acid. Both gave good results, although the more amount of reductants were necessary. These results were added in Table S1 in the supporting information.

  • The statement in lines 327-329 “Most iodoarenes reacted with acrylate esters or acrylamides to give the products in good to excellent yields in the presence of 0.1 mol% 1 and hydrazine regardless of the presence of electron-donating or electron-withdrawing groups on the aromatic ring (entries 6-9)” is not suitable. Only one substrate with methoxy group and ketone was used. Please include a couple more substrates with electron-donating and electron-withdrawing groups.

According to the reviewer’s suggestion, we added a few results on the iodoarenes with an electron-donating group and an electron-withdrawing group for Mizoroki-Heck reactions in Table 3, and one result on iodothiophene for Sonogashira reaction in Table 7. We hope those satisfy the review’s suggestion.

  • The reaction temperature for the Mizoroki-Heck reaction is high. The ppm level of Pd is shown to work in water conditions at rt-45 °C. Does the reaction work at a low temperatures? Similarly, the reaction temperature for Sonogashira coupling is high given that the catalysis in water using Pd-catalyst has been shown to work at lower temperatures i.e., 45 °C with broader substrate scope.

Because we employed thermoresponsive micelle, the micelles form only above 50 °C. So conducting reactions at lower temperature in not very interesting for us. Nevertheless, we tried a reaction at room temperature, but reaction did not proceed at all and the starting materials were recovered. This result was added in Table S1 in supporting information.

  • The substrate scope of the reaction is limited to the aryl substrate. Could you please use some of the heteroaryl-substrate? The is one heteroaryl substrate used for Sonogashira coupling with lower yield i.e., 33 % yield.

The authors thank the reviewer for useful suggestions. We added a few new entries on heteroaryl substrates in Table 4 and 7.